# Cell Free Methylated Tumor DNA in Bronchial Lavage as an Additional Tool for Diagnosing Lung Cancer—A Systematic Review

**DOI:** 10.3390/cancers14092254

**Published:** 2022-04-30

**Authors:** Sara Witting Christensen Wen, Jan Wen, Torben Frøstrup Hansen, Anders Jakobsen, Ole Hilberg

**Affiliations:** 1Department of Oncology, Vejle Hospital, University Hospital of Southern Denmark, Beriderbakken 4, 7100 Vejle, Denmark; torben.hansen@rsyd.dk (T.F.H.); anders.jakobsen@rsyd.dk (A.J.); 2Department of Regional Health Research, J.B. Winsloews Vej 19, 3rd Floor, 5000 Odense C, Denmark; ole.hilberg@rsyd.dk; 3General Practice, Region of Southern Denmark, Damhaven 12, 7100 Vejle, Denmark; jan@wen.dk

**Keywords:** circulating tumor DNA, ctDNA, DNA methylation, lung cancer, bronchial lavage, bronchial wash

## Abstract

**Simple Summary:**

Lung cancer causes the highest number of cancer-related deaths. The prognosis is poor, primarily because the disease is often diagnosed at an advanced stage with no curative treatment options. If lung cancer could be diagnosed at an earlier stage, survival may be improved. DNA is often changed by methylation in cancer cells compared to normal cells. This methylated tumor DNA can be detected in fluid from the lungs, bronchial lavage. Several studies have investigated this biomarker, but the evidence has not been systematically collected. The aim of this review was to identify and synthesize the existing evidence on using methylated tumor DNA in bronchial lavage to diagnose lung cancer. The review will present an overview of the current evidence and contribute to advancing this area further.

**Abstract:**

This systematic review investigated circulating methylated tumor DNA in bronchial lavage fluid for diagnosing lung cancer. PROSPERO registration CRD42022309470. PubMed, Embase, Medline, and Web of Science were searched on 9 March 2022. Studies of adults with lung cancer or undergoing diagnostic workup for suspected lung cancer were included if they used bronchial lavage fluid, analyzed methylated circulating tumor DNA, and reported the diagnostic properties. Sensitivity, specificity, and lung cancer prevalence were summarized in forest plots. Risk of bias was assessed using the QUADAS-2 tool. A total of 25 studies were included. All were case-control studies, most studies used cell pellet for analysis by quantitative PCR. Diagnostic sensitivity ranged from 0% for a single gene to 97% for a four-gene panel. Specificity ranged from 8% for a single gene to 100%. The studies employing a gene panel decreased the specificity, and no gene panel had a perfect specificity of 100%. In conclusion, methylated circulating tumor DNA can be detected in bronchial lavage, and by employing a gene panel the sensitivity can be increased to clinically relevant levels. The available evidence regarding applicability in routine clinical practice is limited. Prospective, randomized clinical trials are needed to determine the further usefulness of this biomarker.

## 1. Introduction

Lung cancer is projected to cause the highest number of estimated cancer-related deaths in the USA in 2022 [1]. The high mortality is mainly due to lung cancer being discovered in an advanced stage with distant metastases in 46% of the cases diagnosed in 2014–2018 [1]. Screening with low-dose computed tomography (CT) scans of the chest has been shown to reduce the mortality in both the National Lung Screening Trial (NLST) in the USA [2] and more recently in the Nederlands-Leuvens Longkanker Screenings Onderzoek (NELSON) in the Netherlands and Belgium [3]. In both studies, the mortality was reduced by more than 20% in the low-dose CT screening group. Since lung cancer screening with low-dose CT was recommended in the USA in 2013, the proportion of cases diagnosed with localized stage disease increased, while the proportion diagnosed with advanced stage disease decreased [1]. 

However, low-dose CT has a high rate of false-positive results. The NLST reported a false-positive rate of 26.6% [2]. This can lead to redundant diagnostic examinations or treatment. Standard diagnostic workup for lung cancer usually includes fiberoptic bronchoscopy with bronchial lavage and ultrasound-guided transbronchial biopsy or transthoracic lung biopsy. These invasive procedures involve a risk of bleeding, infection, and pneumothorax [4,5]. The complication rate may be higher in case of smaller lesions <20 mm [6]. There is a need to improve the diagnostic accuracy of CT-based lung cancer screening in order to reduce the false-positive rate and, consequently, the risk of complications. 

Biomarkers have been suggested as a complementary investigation in lung cancer screening [7]. They may be incorporated in a risk-prediction model along with basic information such as age, smoking status, and pack years [8]. Biomarkers may also hold potential as a diagnostic adjunct to resolve equivocal cytology results [9,10]. Circulating tumor DNA (ctDNA) is one of the frequently investigated cancer biomarkers. It can be detected in the form of oncogenic mutations in genes such as EGFR or KRAS. However, each of the oncogenic mutations is not very prevalent, and tumor heterogeneity must be taken into account [11]. Hence, a large gene panel would be needed for sufficient sensitivity. ctDNA can also be detected as genes with aberrant methylation patterns. Methylation is an epigenetic change caused by adding methyl-groups to the DNA. This prevents transcription of the DNA and thus silencing of the gene [12]. Aberrant DNA methylation has been shown to be a stable change and has been suggested as a biomarker for both diagnostic and prognostic purposes [13,14,15]. 

Biologic materials such as blood, sputum, pleural effusion, and bronchial lavage have been tested in the search for the ideal biomarker medium. Blood has the advantage of being readily available and easily accessible, but small tumors may not always shed a sufficient amount of DNA into the blood stream [16,17]. A specimen collected closer to the tumor may be preferable. Sputum is collected non-invasively, but it may not always be from the lower respiratory tract. Bronchial lavage is a procedure performed during bronchoscopy. A volume of sterile saline solution is instilled into the bronchioles as close to the tumor as possible and then collected. It involves an invasive procedure; however, it is usually well tolerated even by frail patients or patients with COPD [18,19]. 

Much effort has gone into investigating various methylation biomarkers in bronchial lavage and other lung fluids, but to our knowledge this area has not recently been covered by a systematic review. We therefore chose to focus on methylated ctDNA detected in bronchial lavage fluid as an additional diagnostic tool in lung cancer. 

## 2. Materials and Methods

### 2.1. Search Strategy 

The electronic databases PubMed, Embase, Medline, and Web of Science were searched for literature on circulating methylated tumor DNA in bronchial lavage fluid in relation to lung cancer. The research question was divided into three blocks with the titles “Lung cancer”, “Bronchial lavage”, and “Methylated circulating tumor DNA”. We chose to include bronchoalveolar lavage and bronchial wash as well, since these terms are sometimes used interchangeably. The search strategy can be viewed in full detail in the Appendix A. The relevant Medical Subject Headings terms (MeSH terms, PubMed) or Subject Headings (Embase, Medline) were identified for each block. Relevant free-text keywords in each block were then searched separately, and MeSH terms/Subject Headings and keywords were combined with the Boolean operator ‘OR’ to obtain all potentially relevant studies. A search of all three blocks combined with the Boolean operator ‘AND’ was performed to identify the studies relevant for this review. The searches were conducted between 25 February and 9 March 2022. We did not contact any study authors or experts in the field when conducting this review. 

All references were imported into Covidence (Covidence, Melbourne, Australia), and duplicates were automatically removed. The titles and abstracts were screened independently by two reviewers. In case of disagreement, the study was discussed between the main reviewers. If disagreement persisted, the third reviewer had the deciding vote. 

### 2.2. Inclusion and Exclusion Criteria 

The potentially relevant studies were reviewed in full text to evaluate whether they fulfilled the inclusion criteria: (1) adults with lung cancer or undergoing diagnostic workup for suspected lung cancer; (2) bronchial or bronchoalveolar lavage was performed and a fluid sample was collected for analyzing cell free methylated tumor DNA; (3) the reference standard was lung cancer diagnosed by histology or cytology; (4) the outcome was diagnostic sensitivity and specificity or enough information reported to calculate these diagnostic measures. The exclusion criteria were: (1) case-reports, literature reviews, or conference abstracts; (2) studies published in languages other than English; (3) no healthy control group or less than 10 healthy controls included. The full inclusion and exclusion process is illustrated in Figure 1. 

### 2.3. Data Extraction and Quality Assessment

Study characteristics were extracted by one reviewer in Covidence using a piloted data extraction form. Data was collected on title, authors, publication year, country in which the study was performed, study aim, number of cases, number of controls, type of biological specimen including whether pellet or supernatant was used, analysis method, and name of the analyzed genes. Study outcomes in the form of sensitivity and specificity for the genes and for potential gene panels were extracted independently by two reviewers. Disagreements were solved as described under Search strategy. The quality of the included studies was assessed by one reviewer using the Quality Assessment of Diagnostic Accuracy Studies 2 (QUADAS-2) tool [20]. 

### 2.4. Data Synthesis 

Data was synthesized in the form of a summary table containing all included studies. If diagnostic sensitivity or specificity was not reported, these measures were calculated as follows. Sensitivity = true positive/(true positive + false negative). Specificity = true negative/(true negative + false positive). Forest plots of sensitivity, specificity, and prevalence were produced in Microsoft Excel (Microsoft Corporation, Redmond, Washington, DC, USA). 

The study was registered in Prospero (registration number CRD42022309470, University of York, York, United Kingdom) on 9 February 2022. All items are reported according to the Preferred Reporting Items for Systematic Reviews and Meta-Analyses of Diagnostic Test Accuracy (PRISMA-DTA) checklist [21].

## 3. Results

### 3.1. Study Characteristics 

The literature search resulted in 1039 studies, which were screened by title and abstract, and 38 studies were reviewed in full text. Eventually, 25 studies were included, and the study characteristics can be seen in Table 1. Of the 9 Asian studies, 6 studies originated from China, while 5 of the 14 European studies were performed in Germany. All studies were designed as case-control studies. The study cohorts varied in size: 10 cohorts had less than 100 subjects in total; 12 cohorts included up to 200 subjects; four cohorts included up to 300 subjects; and three cohorts included >300 subjects. 

A total of 12 out of the 25 studies employed bronchoalveolar lavage as the method for collecting biological material, but both bronchial lavage and bronchial wash were frequently used methods. Cell pellet/precipitate was predominantly used for the analyses; only one study used the supernatant, and two studies used the whole fluid. This aspect of the methods section was not reported in eight studies. Quantitative methylation specific PCR (QMSP) dominated the analysis methods, but three of the more recent studies used droplet digital PCR (ddPCR), and one study used next-generation sequencing (NGS). Four studies divided their patients into test and validation cohorts to establish and validate a specific cut-off, while others used the study to set the cut-off to be validated in a future study. Four studies used a cut-off defined in a previously published work. Five studies did not report how the cut-off distinguishing between cancer vs. benign was chosen. Sampling methods, specimens, methods, and cut-off determination can be viewed in Table 2 for all the included studies. 

### 3.2. Quality of Included Studies 

Study quality according to the QUADAS-2 tool can be viewed in Table 3. There was low risk of applicability concerns in all included studies. We investigated whether ctDNA in bronchial lavage or similar fluids can be used as a diagnostic adjunct in lung cancer. Therefore, studies which did not investigate lung cancer, had no control group, or which did not use a methylated ctDNA biomarker were excluded. The studies must also investigate the diagnostic abilities of the biomarker(s) in order to be included. Most studies had one or more items with unclear risk of bias because key aspects were not reported in enough detail to assess the study quality. This was most frequent in the areas ‘Patient selection’ and ‘Index test’. It was often with regard to whether the patients were selected randomly or consecutively included, and whether the index test results were evaluated blinded to the results of the reference standard. Five studies did not address conflicts of interest [22,25,26,27,32]. One study did not report any conflicts of interest, but the study was performed in order to gain a CE marking for the ctDNA assay [29]. The detailed quality assessments can be viewed in the Appendix A (Appendix A). 

### 3.3. Diagnostic Properties of Methylated Circulating Tumor DNA in Bronchial Lavage Fluid 

The sensitivity and specificity of methylated circulating tumor DNA for diagnosing lung cancer differed greatly between studies. Diagnostic sensitivity ranged from 0% for a few single genes [24,25] to 97% for a four-gene panel [41]. Specificity ranged from 8% for a single gene [27] to 100% in several studies [24,26,27,30,41]. Generally, the studies which increased sensitivity by employing a gene panel also decreased the specificity, and no combined gene panel had a perfect specificity of 100%. Figure 2 illustrates the diagnostic properties of the combined biomarker panels or the best performing single gene from each included study. The whole range of sensitivity and specificity for all genes from the included studies can be viewed in the Appendix A (Appendix A). 

The most frequently investigated single genes were p16(INK4A) and RASSF1A; each were investigated in seven studies. Specificity was generally high, while p16 sensitivity ranged from 12–24% (Figure 3A) and RASSF1A sensitivity ranged from 18–51% (Figure 3B). Forest plots of the second most frequently investigated genes, SHOX2 and RARB2, can be viewed in the Appendix A (Appendix A). 

## 4. Discussion

This review identified 25 independent studies addressing the question of whether methylated ctDNA in bronchial lavage fluid is relevant as an additional diagnostic tool in lung cancer. The available data suggest that this type of biological specimen may be a relevant medium for detecting ctDNA with a sensitivity >50% in the majority of the included studies. The diagnostic properties depended largely on the gene(s) chosen for investigation and on the analysis methods. 

The included studies were mainly European or Asian with only one American [24] and one Egyptian [40] study. All studies were of a case-control design. We did not identify any prospective interventional studies aiming to assess the diagnostic properties of methylated ctDNA in bronchial lavage in a randomized manner. This was as expected since this approach is still relatively new. Many ctDNA assays need to be validated and standardized [45], and only the Epi proLung BL Reflex assay^®^ is commercially available and has received the CE-IVD mark [46]. There are not yet any methylation-based biomarkers approved by the United States Food and Drug Administration [47]. 

One of the important pre-analytical factors is the choice of biological specimen. The studies included in this review used both bronchoalveolar lavage, bronchial lavage, and bronchial wash as approaches to sample collection. The majority of the studies utilized the pellet/precipitate for the ctDNA analysis. This could be advantageous because the pellet contains tumor cells with larger amounts of tumor DNA. On the other hand, the pellet likely contains a large number of normal cells such as epithelial cells. The large amounts of normal DNA could potentially interfere in the ctDNA analyses and their interpretation. One way to overcome this obstacle is to report the relative instead of the absolute differences in methylated ctDNA [48]. This method or similar methods for relative methylation expression was used by 13 of the included studies, which analyzed ctDNA by quantitative methylation specific PCR or ddPCR. Three studies used either supernatant [43] or whole fluid [28,36], and their results were quite similar to the rest of the included studies utilizing the cell pellet. This leads to the tentative conclusion that the whole lavage fluid or supernatant may be as useful as the pellet for detecting methylated ctDNA. However, this needs to be formally investigated in a comparative study. 

The genes p16(INK4A) and RASSF1A were the two most frequently investigated genes, and the specificity was generally very high. The sensitivity, however, was maximally 51% [30], which was on the lower end of the spectrum compared to the sensitivity accomplished by the combined gene models. This could be because p16(INK4A) and RASSF1A were primarily investigated by the older studies. The most recent study analyzing p16(INK4A) was Nikolaidis 2012 [30], while RASSF1A was analyzed more recently by Ren 2017 [33] and Roncarati 2020 [41]. Older studies [22,23] may employ less-sensitive analysis methods compared to, e.g., digital PCR [41]. This may also explain the very high specificity achieved. Generally, sensitivity was increased by implementing a panel of several genes with the criterion of at least one aberrantly methylated gene equaling a positive test. However, this also tended to decrease specificity since it also increased the risk of false positive tests. Another approach could be to integrate different types of biomarkers such as host immunologic factors or protein markers [49,50]. A four-protein biomarker panel was reported to significantly improve a lung cancer prediction model [51]. It would be interesting to combine a methylation gene panel with protein markers and clinical risk factors in a multi-factorial model. 

In total, 17 of the included studies did not report important aspects of their patient selection or analysis methods (Table 3). Missing information in these important areas makes it difficult to reproduce the studies and evaluate the study quality. Guidelines such as the Standards for Reporting of Diagnostic Accuracy Studies (STARD) [52] criteria are very useful in ensuring that all essential information is reported. This is crucial for the further development and standardization of ctDNA diagnostic methods in order for them to be applied in routine clinical practice. 

The present review has several limitations. We employed a wide search strategy, but we might have missed relevant reports. We did not include studies in languages other than English. As we discovered at least one article in Chinese, which was excluded, we may have missed other important studies. We were not able to formally test for or evaluate the risk of publication bias, because most of the studies did not report standard errors or 95% confidence intervals with their diagnostic measures. All of the included studies reported at least one statistically significant result favoring the ctDNA methylation markers for diagnosing lung cancer. This may be caused by publication bias towards the publishing of positive results. The study characteristics and the study quality were only assessed by one reviewer, which increases the risk of errors. The outcomes, however, were extracted by two independent reviewers as recommended by the Cochrane Handbook of Systematic Reviews [53]. 

## 5. Conclusions

In conclusion, methylated ctDNA can be detected in bronchial lavage or similarly obtained fluids, and by employing a gene panel the sensitivity can be increased to clinically relevant levels. However, the available evidence regarding applicability in routine clinical practice is limited. Prospective, randomized clinical trials are needed in order to determine the further usefulness of this diagnostic biomarker. 

## Figures and Tables

**Figure 1 cancers-14-02254-f001:**
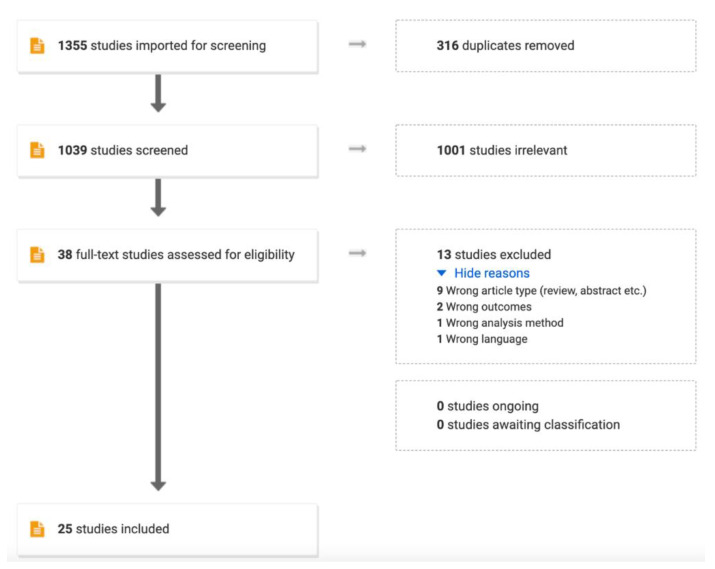
PRISMA flowchart illustrating the process from screening to inclusion of the studies for the review.

**Figure 2 cancers-14-02254-f002:**
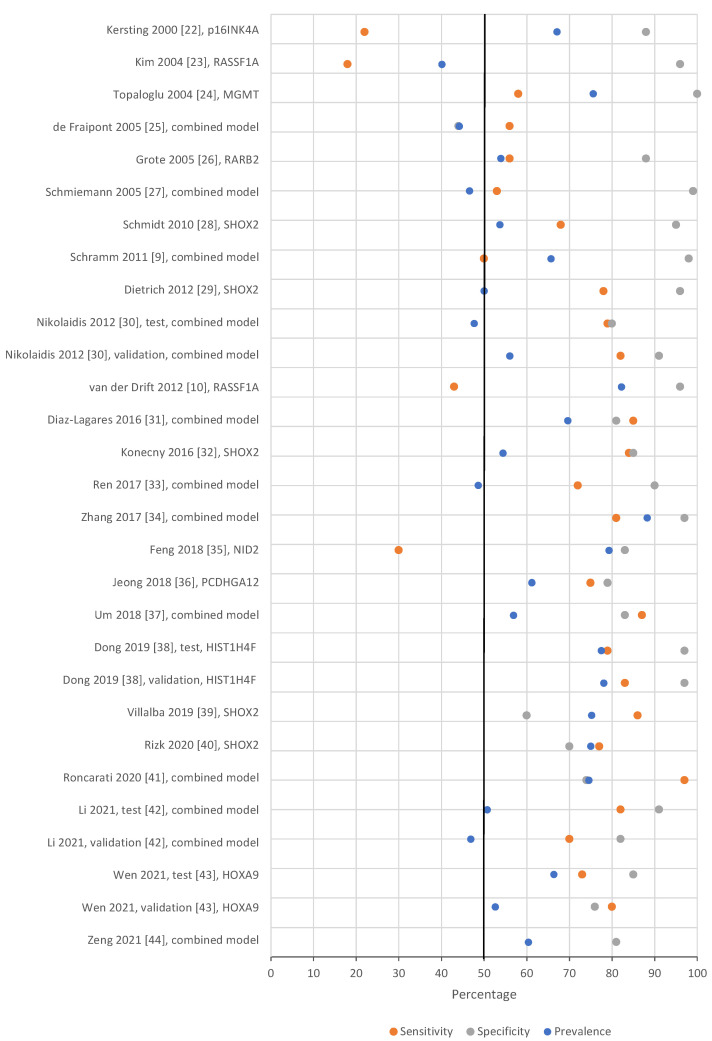
Forest plot of sensitivity (orange), specificity (gray), and prevalence of lung cancer (blue) for the combined biomarker panels or the best performing single gene from each included study. If a test and validation approach was used, both cohorts were included in the graph. The x-axis represents study sensitivity, specificity, and prevalence in percent. The vertical, black line represents the 50% mark. There are no whiskers, since many studies did not report a 95% confidence interval, standard error, or similar error margins [9,10,22,23,24,25,26,27,28,29,30,31,32,33,34,35,36,37,38,39,40,41,42,43,44].

**Figure 3 cancers-14-02254-f003:**
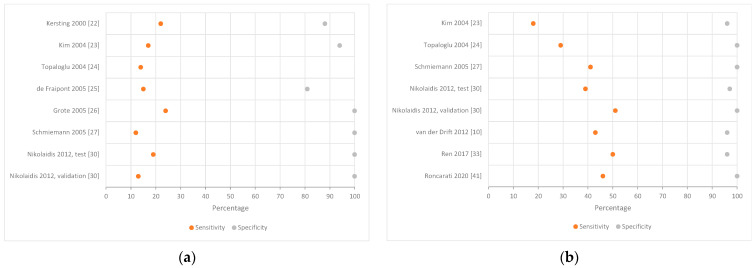
Forest plot illustrating the sensitivity (orange) and specificity (gray) of the most frequently investigated genes (**a**) p16(INK4A) and (**b**) RASSF1A. The x-axis represents study sensitivity and specificity in percent. There are no whiskers, since many studies did not report a 95% confidence interval [10,22,23,24,25,26,27,30,33,41].

**Table 1 cancers-14-02254-t001:** Four studies were divided into test and validation cohort, and one study had separate cohorts for bronchoalveolar lavage (BAL) and bronchial aspirates. The case description ‘All types’ covers both small-cell lung cancer and non-small-cell lung cancer (NSCLC). The case description ‘All stages’ covers the tumor, node, and metastasis (TNM) stages I–IV.

Study ID	Country	Study Design	Cases, n	Cases, Description	Controls, n	Controls, Description
Kersting 2000 [22]	Germany	Case-control	51	All types, all stages	25	Symptomatic smokers >20 pack years, no current lung cancer
Kim 2004 [23]	Korea	Case-control	85	Surgically resected NSCLC	127	No current or historic malignancies
Topaloglu 2004 [24]	USA	Case-control	31	NSCLC, all stages	10	Age-matched, no current lung cancer
de Fraipont 2005 [25]	France	Case-control	34	Primary and previously operated NSCLC	43	No current lung cancer
Grote 2005 [26]	Germany	Case-control	75	All types, all stages	64	No current lung cancer
Schmiemann 2005 [27]	Germany	Case-control	89	All types, all stages	102	No current lung cancer
Schmidt 2010 [28]	Germany, England	Case-control	281	All types, all stages	242	No current lung cancer
Schramm 2011 [9]	Germany	Case-control	117	All types, all stages	61	No current or historic lung cancer
Dietrich 2012 [29]	England	Case-control	125	All types, stages unkown	125	No current malignancy
Nikolaidis 2012 [30]	England	Case-control	Test: 194 Validation: 139	All types, all stages	Test: 213 Validation: 109	No current lung cancer; 36 patients with other cancers
van der Drift 2012 [10]	The Netherlands	Case-control	129	All types, all stages	28	No current lung cancer
Diaz-Lagares 2016 [31]	Spain	Case-control	51 aspirates82 BAL	All types, all stages	29 aspirates29 BAL	No current lung cancer
Konecny 2016 [32]	Slovakia	Case-control	37	All types, all stages	31	No current lung cancer
Ren 2017 [33]	China	Case-control	123	All types, all stages	130	No current lung cancer; 18 patients with other cancers
Zhang 2017 [34]	China	Case-control	284	All types, all stages	38	No current lung cancer; 3 patients with other cancers
Feng 2018 [35]	China	Case-control	46	NSCLC, all stages	12	No current lung cancer
Jeong 2018 [36]	Korea	Case-control	60	All types, all stages	38	No current lung cancer
Um 2018 [37]	Korea	Case-control	70	NSCLC stage I-IIIa	53	No current lung cancer
Dong 2019 [38]	China	Case-control	Test: 103 Validation: 103	NSCLC, all stages	Test: 30 Validation: 29	No current lung cancer
Villalba 2019 [39]	Spain	Case-control	79	NSCLC, all stages	26	No current lung cancer
Rizk 2020 [40]	Egypt	Case-control	60	NSCLC, stages unknown	20	Sex and age matched with no current lung cancer
Roncarati 2020 [41]	Italy	Case-control	91	All types, all stages	31	No current lung cancer
Li 2021 [42]	China	Case-control	Test: 36 Validation: 52	NSCLC, all stages	Test: 35 Validation: 59	No current lung cancer
Wen 2021 [43]	Denmark	Case-control	Test: 67 Validation: 50	All types, all stages	Test: 34 Validation: 45	No current lung cancer
Zeng 2021 [44]	China	Case-control	32	Solid nodule < 2 cm	21	No current lung cancer

**Table 2 cancers-14-02254-t002:** Sampling methods: Bronchial lavage (BL), bronchoalveolar lavage (BAL), bronchial wash (BW), and bronchial aspirates (BA). Specimen used: Cell pellet, supernatant, or unprocessed or fixed fluid samples. Analysis methods: Non-quantitative polymerase chain reaction (PCR), quantitative methylation specific PCR (QMSP), droplet digital PCR (ddPCR), chip/microarray, pyrosequencing, Sanger sequencing, or next-generation sequencing (NGS). Cut-off: Analysis not quantitative, cut-off defined in a previous study, receiver operating characteristics (ROC) analysis of the present study or in a test and validation set-up.

Study ID	Sampling Method	Specimen	Method(s)	Cut-Off
Kersting 2000 [22]	BL	Pellet	PCR, non-quantitative	Not quantitative.
Kim 2004 [23]	BL	Pellet	PCR, non-quantitative	Not quantitative.
Topaloglu 2004 [24]	BAL	Pellet	QMSP	The highest methylation found in three normal controls was set as the cut-off for the case samples.
de Fraipont 2005 [25]	BL	Pellet	QMSP	Not reported.
Grote 2005 [26]	BW and BAL	Not reported	QMSP	A cutoff of >30% methylation for RARB2 was defined in the study.
Schmiemann 2005 [27]	BW and BAL	Not reported	QMSP	Defined in a previous study.
Schmidt 2010 [28]	BA	Pellet from the unfixed samples, whole fluid from the Saccomanno fixed samples.	QMSP Chip/microarray	The cutoff that resulted in <5% false positive rate in the benign samples.
Schramm 2011 [9]	BW	Pellet	QMSP	Defined in a previous study.
Dietrich 2012 [29]	BL	Pellet	QMSP	Defined in a previous study.
Nikolaidis 2012 [30]	BL	Pellet	QMSP	Defined by a test cohort using ROC analysis.
van der Drift 2012 [10]	BW	Pellet	QMSP	Not reported.
Diaz-Lagares 2016 [31]	BA and BAL	Not reported	Pyrosequencing Chip/microarray	Defined by a test cohort using ROC analysis.
Konecny 2016 [32]	BL	Pellet	QMSP	Defined in a previous study.
Ren 2017 [33]	BAL	Pellet	QMSP Sanger sequencing	Not reported.
Zhang 2017 [34]	BAL	Pellet	QMSPSanger sequencing	Not reported.
Feng 2018 [35]	BAL	Pellet	QMSP	Not reported.
Jeong 2018 [36]	BW	3–5 mL of the fluid, presumably unprocessed.	QMSP	ROC analysis in the present study, no validation.
Um 2018 [37]	BW	Not reported	Chip/microarray Pyrosequencing	Defined by a test cohort using ROC analysis.
Dong 2019 [38]	BAL	Not reported	QMSP Pyrosequencing	Defined by a test cohort using ROC analysis.
Villalba 2019 [39]	BAL	Not reported	ddPCR	ROC analysis in the present study, no validation.
Rizk 2020 [40]	BAL	Not reported	QMSP	ROC analysis in the present study, no validation.
Roncarati 2020 [41]	BW	Pellet	ddPCR	Poisson distribution to quantify absolute number of droplets. Sample considered positive when both duplicate experiments were positive.
Li 2021 [42]	BAL	Pellet	QMSP	Defined by a test cohort using ROC analysis.
Wen 2021 [43]	BL	Supernatant	ddPCR	Defined by a test cohort using ROC analysis.
Zeng 2021 [44]	BAL	Not reported	NGS	From analyzing tissues.

**Table 3 cancers-14-02254-t003:** Risk of bias assessed by the QUADAS-2 tool.

Study ID	Risk of Bias				Applicability Concerns		
	Patient Selection	Index Test	Reference Standard	Flow and Timing	Patient Selection	Index Test	Reference Standard
Kersting 2000 [22]	Low risk	Unclear	Low risk	Low risk	Low risk	Low risk	Low risk
Kim 2004 [23]	Low risk	Unclear	Low risk	Low risk	Low risk	Low risk	Low risk
Topaloglu 2004 [24]	Unclear	High risk	Low risk	Low risk	Low risk	Low risk	Low risk
de Fraipont 2005 [25]	Unclear	Unclear	Unclear	Low risk	Low risk	Low risk	Low risk
Grote 2005 [26]	Low risk	Low risk	Low risk	Low risk	Low risk	Low risk	Low risk
Schmiemann 2005 [27]	Low risk	Low risk	Low risk	Low risk	Low risk	Low risk	Low risk
Schmidt 2010 [28]	Unclear	Unclear	Low risk	Low risk	Low risk	Low risk	Low risk
Schramm 2011 [9]	Low risk	Low risk	Low risk	Low risk	Low risk	Low risk	Low risk
Dietrich 2012 [29]	High risk	High risk	Low risk	Low risk	Low risk	Low risk	Low risk
Nikolaidis 2012 [30]	High risk	Unclear	Low risk	Low risk	Low risk	Low risk	Low risk
van der Drift 2012 [10]	Unclear	Unclear	Unclear	Low risk	Low risk	Low risk	Low risk
Diaz-Lagares 2016 [31]	High risk	Unclear	Low risk	Low risk	Low risk	Low risk	Low risk
Konecny 2016 [32]	Unclear	Unclear	Unclear	Low risk	Low risk	Low risk	Low risk
Ren 2017 [33]	Unclear	Unclear	Low risk	Low risk	Low risk	Low risk	Low risk
Zhang 2017 [34]	High risk	High risk	Low risk	Low risk	Low risk	Low risk	Low risk
Feng 2018 [35]	Unclear	Unclear	Unclear	Low risk	Low risk	Low risk	Low risk
Jeong 2018 [36]	Low risk	Unclear	Unclear	Low risk	Low risk	Low risk	Low risk
Um 2018 [37]	High risk	Low risk	Low risk	Low risk	Low risk	Low risk	Low risk
Dong 2019 [38]	Unclear	Unclear	Unclear	Unclear	Low risk	Low risk	Low risk
Villalba 2019 [39]	Unclear	High risk	Low risk	Low risk	Low risk	Low risk	Low risk
Rizk 2020 [40]	Unclear	Unclear	Unclear	Low risk	Low risk	Low risk	Low risk
Roncarati 2020 [41]	Low risk	Unclear	Low risk	Low risk	Low risk	Low risk	Low risk
Li 2021 [42]	Unclear	Unclear	Low risk	Low risk	Low risk	Low risk	Low risk
Wen 2021 [43]	Low risk	Low risk	Low risk	Low risk	Low risk	Low risk	Low risk
Zeng 2021 [44]	Low risk	Low risk	Low risk	Low risk	Low risk	Low risk	Low risk

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
