# Peer review of "Cell Free Methylated Tumor DNA in Bronchial Lavage as an Additional Tool for Diagnosing Lung Cancer—A Systematic Review"

_cancers, 2022, doi:10.3390/cancers14092254_

Round 1
Reviewer 1 Report
An interesting review is presented for the review, which undoubtedly deserves the attention of readers. There are several questions/comments to the authors: 1) I did not see information about what histological type of lung cancer is in the studies included in the review, there is no information about the stage of the disease and the presence/absence of distant and regional metastasis. This information may help in interpreting the results. 2) Has there been a parallel determination of methylated tumor DNA in the blood in studies? If yes, then I would like to understand whether there is a correlation with blood or not.
Author Response
Thank you for taking the time to review our work. We very much appreciate your comments. Please see below for the replies to your comments.
1) We did not report the histological types of lung cancer or the stages included in the selected studies. We have now added a short description of the cases and controls in Table 1. This information includes whether the study reported all types of lung cancer or only non-small cell lung cancer and whether all stages or specific disease stages were reported. The reviewer is correct in pointing out that this information may help readers interpret the results.
2) Some of the studies determined methylated tumor DNA in blood in parallel with bronchial lavage or similar lung fluids. These results and the comparison between the specimens are very interesting and important in order to further the understanding of ctDNA. Yet, we believe that this comparison is outside the scope of the present work. We do, however, think that the comparison between methylated ctDNA in blood compared to bronchial lavage should be addressed in a separate review.
Reviewer 2 Report
Recently I was invited to review an interesting paper entitled: "Cell free methylated tumor DNA in bronchial lavage as an additional tool for diagnosing lung cancer – A systematic review". Advances in the precise assessment of the prognosis and the treatment plan gain significant attention in the scientific and clinical community of specialists interested in NSCLC. It is important to evaluate the circulating factors that can help to optimize the therapy of patients with NSCLC. Bronchial lavage may be one of the methods to obtain valuable samples that may allow for the analysis of circulating tumor DNA. The authors carefully present the systematic review. I find it interesting and well-written. I find no major flaws. The paper is prepared maturely. It carries a significant educational and scientific load. I recommend publication without further revisions.
Author Response
Thank you for taking the time to review our work. Your comments were very kind, and we appreciate your help.
Reviewer 3 Report
In this review manuscript, the authors selected 25 independent studies and wanted to find out whether methylated ctDNA from bronchial lavage fluid from lung cancer patients can be used as a diagnostic tool to screen for lung cancer. The followings are some concerns and comments have been pointed out that the authors may want to consider.
- The early screening of lung cancer is very important to prolong the survival of cancer patients. The authors stated “This review identified 25 independent studies addressing the question of whether methylated ctDNA in bronchial lavage fluid is relevant as an additional diagnostic tool in lung cancer” lines 229-231 and lines 21-22. Please confirm, did the authors care about the early-stage screening of lung cancer or not? If yes, I’d suggest the authors point out which selected studies include the related information.
- Lines 164-166: Within the same sentence, “ten cohorts”, “12 cohorts”, “four cohorts”, and “three cohorts”, please homogenous the format of the number.
- Line 169, line 245, line 251, line 270, and line 282: Please specify “majority of the studies” and “many of the included studies”. For example, how many studies out of 25 selected studies and whatever method the authors preferred to make them clearer.
- Table 2, Table 3, Table S1, Table S2, Table S3, Table S4, Table S5, and Table S6, Figure 2: Please add the reference serial number to “study ID” for easier tracking.
- Table 2, Table 3, Table S1, Table S2, Table S3, Table S4, Table S5, and Table S6, should be mentioned and cited in the main context.
- Please use italic p as it refers to a p-value. Check throughout the manuscript.
- Line 190: I’d suggest the authors emphasize the review question here.
- Line 199: Please specify which part/section of supplementary materials.
- Line 205: It seems there is an extra space before the word “to”. Please check.
- Line 206: Please specify “several studies”.
- Line 211: Please specify which part/section of supplementary materials.
- Line 213 Figure 2 and line 224 Figure 3: Please add a color legend in the image instead of only in the figure legend by text. Add X-axis legend under or next to X-axis as well for easier tracking.
- Line 222: Please specify which part/section of supplementary materials.
- Line 232: Please specify “>50%” and “majority” to make it clearer and easier tracking and understandable by readers. For example, bold the study ID in Table 2 or whatever method that the authors preferred.
- Line 254: Please specify “other included studies”.
- Line 261: Please specify “the older studies”. For example, list some references, etc.
- Lines 267-268: Provide literature “A four-protein biomarker panel was reported to significantly improve a lung cancer prediction model”.
- References: Please cite the literature correctly, for example, line 327 reference 18 should be cited in the following format instead of the current style in the manuscript: /Oudkerk M, Liu S, Heuvelmans MA, Walter JE, Field JK. Lung cancer LDCT screening and mortality reduction - evidence, pitfalls and future perspectives. Nat Rev Clin Oncol. 2021 Mar;18(3):135-151. doi: 10.1038/s41571-020-00432-6. Epub 2020 Oct 12. PMID: 33046839./. Please check all the other citations.
- Lines 405-406 reference 41: “Mol Oncol. 2020 May 22;1878-0261.12713.” could not hit the titled reference, please double check and consider the following citation information for this literature /Roncarati R, Lupini L, Miotto E, Saccenti E, Mascetti S, Morandi L, Bassi C, Rasio D, Callegari E, Conti V, Rinaldi R, Lanza G, Gafà R, Papi A, Frassoldati A, Sabbioni S, Ravenna F, Casoni GL, Negrini M. Molecular testing on bronchial washings for the diagnosis and predictive assessment of lung cancer. Mol Oncol. 2020 Sep;14(9):2163-2175. doi: 10.1002/1878-0261.12713. Epub 2020 Jun 24. PMID: 32441866; PMCID: PMC7463327./
Author Response
Thank you for taking the time to review our work. We very much appreciate your comments. Please see below for the replies to your comments.
1) We certainly care about early-stage screening for lung cancer, but unfortunately, we neglected to include the disease stages, which were reported in each study. We have now added a short description of the cases and controls in Table 1. This information includes whether the study reported all types of lung cancer or only non-small cell lung cancer and whether all stages or specific disease stages were reported.
2) We usually spell out numbers from one to nine, but when starting a sentence with a number of ten or more, we usually spell it out as well. I hope that this system is acceptable to you, but if you prefer, we will change it upon request.
3) Yes, you are correct. The phrasing was unclear in several places. We have tried to make the points clearer.
4) Thank you for the good suggestion, we have added the references to all the relevant tables and figures.
5) Yes, of course. They have all been cited in the text now.
6) Thank you, this has been corrected.
7) We have incorporated the review question here. Please see lines 295-299.
8) We have referenced Table S1-S5 accordingly.
9) Yes, sorry for mistyping.
10) We have added references for the statement (line 336).
11) Table S6 has now been referenced.
12) Figures 2 and 3A and 3B have been updated accordingly. Thank you for the excellent suggestions.
13) We have now referenced Figure S1 and S2 in the text.
14) We have added a line at the 50% mark in Figure 2 to make it more visually clear, that the majority of the dots ( = the majority of the sensitivity and specificity points) are >50%.
15) This simply refers to the rest of the studies included in the review. We have tried to make this easier to understand. Please see lines 397-399.
16) We have now elaborated and specified. Please see lines 409-412.
17) That reference was forgotten by mistake. It has been corrected.
18) We have used the version published online ahead of print, but we have now updated the reference. Thank you.
19) We did not have any troubles locating the article from the cited reference, but we have updated the reference on the reviewer’s request. We have double checked and corrected the reference list to include the doi-numbers.
Round 2
Reviewer 1 Report
I have no more remarks/comments on the article. I think that in its present form the article can be recommended for publication.
Author Response
Thank you again for taking the time to review our manuscript, we very much appreciate it.
Reviewer 3 Report
I don’t have many concerns about the current manuscript now except for the following that the authors should consider before publication.
1: Please make sure Figure 1 is a high resolution image before publication.
2: Please seriously revise the manuscript based on all the previous comments from editors and reviewers, and so on before publication.
3: Please format the reference with /Cancers/ requirement.
/In the text, reference numbers should be placed in square brackets [ ], and placed before the punctuation; for example [1], [1–3] or [1,3]. https://www.mdpi.com/journal/cancers/instructions /
Author Response
Thank you for taking the time to review our manuscript for the second time. We have tried to comply with all of the comments to our best efforts. Please see point by point below.
1: Figure 1 has a resolution of 1699 × 1351 Pixels (JPEG file). I hope that is sufficient.
2: We have done our utmost to comply with all comments and suggestions from reviewers and editors. Please make further comments or suggestions, if you find anything else which ought to be corrected.
3: We have now put all references in square brackets.